# Trends in Nursing Research on Infections: Semantic Network Analysis and Topic Modeling

**DOI:** 10.3390/ijerph18136915

**Published:** 2021-06-28

**Authors:** Jongsoon Won, Kyunghee Kim, Kyeong-Yae Sohng, Sung-Ok Chang, Seung-Kyo Chaung, Min-Jung Choi, Youngji Kim

**Affiliations:** 1College of Nursing, Eulji University, Seongnam-si 13135, Korea; jswon@eulji.ac.kr; 2Red Cross College of Nursing, Chung-Ang University, Seoul 06974, Korea; kyung@cau.ac.kr; 3College of Nursing, The Catholic University of Korea, Seoul 06591, Korea; sky@catholic.ac.kr (K.-Y.S.); genius0527@naver.com (M.-J.C.); 4College of Nursing, Korea University, Seoul 02841, Korea; sungok@korea.ac.kr; 5Department of Nursing, Semyung University, Jecheon-si 27136, Korea; chaungck@hanmail.net; 6College of Nursing and Health, Kongju National University, Gongju-si 32588, Korea

**Keywords:** infection, nurses, semantics, text mining, nursing

## Abstract

Background: Many countries around the world are currently threatened by the COVID-19 pandemic, and nurses are facing increasing responsibilities and work demands related to infection control. To establish a developmental strategy for infection control, it is important to analyze, understand, or visualize the accumulated data gathered from research in the field of nursing. Methods: A total of 4854 articles published between 1978 and 2017 were retrieved from the Web of Science. Abstracts from these articles were extracted, and network analysis was conducted using the semantic network module. Results: ‘wound’, ‘injury’, ‘breast’, “dressing”, ‘temperature’, ‘drainage’, ‘diabetes’, ‘abscess’, and ‘cleaning’ were identified as the keywords with high values of degree centrality, betweenness centrality, and closeness centrality; hence, they were determined to be influential in the network. The major topics were ‘PLWH’ (people living with HIV), ‘pregnancy’, and ‘STI’ (sexually transmitted infection). Conclusions: Diverse infection research has been conducted on the topics of blood-borne infections, sexually transmitted infections, respiratory infections, urinary tract infections, and bacterial infections. STIs (including HIV), pregnancy, and bacterial infections have been the focus of particularly intense research by nursing researchers. More research on viral infections, urinary tract infections, immune topic, and hospital-acquired infections will be needed.

## 1. Introduction

Many countries around the world are currently threatened by the COVID-19 pandemic. As of mid-December 2020, 1,630,000 people have died from COVID-19 and more than 73 million people have been infected [1]. This marks the third crisis in the 21st century caused by the spread of a novel respiratory virus—after the SARS pandemic in 2002 and the MERS outbreak in 2015—wherein medical institutions have experienced a rapid increase in patients with an emerging infectious disease.

The risk of transmission of infections to the community at medical institutions is always present because medical institutions have many employees and visitors, as well as patients with infectious diseases and frail patients susceptible to infections. Therefore, it is very important to establish and implement strategies to protect patients, visitors, employees, and community members from infections [2].

Accordingly, medical institutions and health authorities are striving to establish infection management systems to protect medical institutions and communities from the increasing threat of infection. As nurses are key health personnel in charge of infection prevention and management, they are at the forefront of these initiatives; therefore, nurses are facing increasing responsibilities and work demands related to infection control [3,4].

In this context, an in-depth exploratory study would shed valuable light on whether related research is being conducted in connection with the increased risk of infection, and the findings of such an analysis would have important implications for establishing infection control strategies. Therefore, the research trend of infection research in the field of nursing should be characterized. Although it is known that extensive research related to infection in the field of nursing has been carried out, it is more challenging to understand what insights and value are conveyed by the large amount of published data [5]. For this reason, an analysis method is needed to extract information by searching large amounts of data and organizing it in a way that helps researchers to identify the patterns hidden in big data.

Text network analysis visualizes text from big data sources, which contain a wide range of text data, as a network, showing the most relevant keywords, as well as their relationships and structures [6]. The contextual relationships of the text can be grasped by using social network analysis indicators, such as Freeman’s three centrality indicators (degree centrality, betweenness centrality, and closeness centrality) [7]. Topic modeling is a probabilistic algorithm that extracts themes (topics) from a large amount of text data to obtain insights and new ideas using computer-aided content analysis techniques [8]. The usefulness of these analytic methods has been verified, and they have recently been used in several academic fields, including nursing.

The aim of this study was to identify the core keywords and topics of infection-related research over the past 40 years to better understand research trends in the field of nursing and public health. These findings can serve as a cornerstone for strategic developmental directions in infection control and may furnish directions for future research.

## 2. Materials and Methods

### 2.1. Study Design and Research Procedure

This study was a quantitative content analysis to identify the core keywords and to explore topics of research on infections in the academic field of nursing. Abstracts from published infection studies constituted the dataset. The relationship between concurrent keyword emergences was identified to investigate infection research trends and characteristics of the relevant field.

### 2.2. Data Collection and Analysis


(1)Selection of literature for analysis: The years of publication for articles were limited to between 1978 and 2017, and a literature search was conducted from January to April 2018 using Web of Science. Seven researchers made a list of relevant search keywords by referring to MeSH terms and previous literature reviews, and established the final search keywords through a discussion. The following keywords were used as search terms: ‘infection’, ‘disinfection’, ‘protective isolation’, ‘standard precaution’, ‘surgical asepsis’, ‘medical asepsis’, ‘infectious disease’, ‘respiratory protection’, ‘personal protective equipment’, ‘isolation’, ‘isolation precaution’, ‘sterilization’, ‘apron’, ‘gloves or gloving’, ‘gowns or gowning’, ‘hand hygiene’, ‘hand washing’, ‘healthcare associated infection’, ‘hospital infection’, and ‘nosocomial infection’. One author (JW) compiled a comprehensive dataset from the search results.(2)Selection of the analysis units: In total, 7082 articles were retrieved from Web of Science, which includes a comprehensive range of nursing, medical, psychological, and social scientific literature. The retrieved articles were equally divided and randomly distributed by one author (KK) among seven researchers for a visual examination of the abstract for every record. After reviewing the distributed articles, all seven researchers jointly discussed the content, and after refinement of the results through discussion and consensus, a total of 4854 papers were selected for analysis, after the exclusion of 2228 papers that were not related to nursing, were unsuitable for the research topic, or did not have the appropriate format of a paper. Even if an article was not written in English, it was included in the dataset if an English abstract was provided. When searching the literature, the country was not designated separately, and only journals were included in the article format. Conference abstracts and reports by organizations were excluded. The original search results contained articles related to infection in various fields, including animal studies and physiology, while the refined database used for further analysis was composed of literature in the field of nursing science.(3)Generation of co-occurrence matrices and network: The abstracts of the selected documents were extracted and arranged into a single row in Excel, which was exported into the NetMiner text mining program. The keywords used in the abstracts were extracted using a semantic network module in the NetMiner program (NetMiner v 4.4, Cyram Co. Ltd., Gyeonggi-do, Korea). All nouns were collected by auto-filtering in this program and were manually converted to singular forms, and synonyms were grouped together and expressed by a representative word [9]. The user dictionary of this program enabled us to register proper nouns, compound nouns, and newly coined words. It also allowed us to use a single word to register words with similar meanings. The terms were classified by two researchers into major terms, which were directly related to infection, and minor terms, which were not directly related. There was a high rate of agreement (approximately 88%). Disagreements were resolved through consensus. A thesaurus made by the authors was generated through the literature review, and the terms were refined using the dictionary function of the software. The keywords were arranged focusing mainly on nouns, and medical terms composed of several words referring to a disease, test, or instrument were treated as one word (Table 1). Two researchers reviewed the thesaurus separately and disagreements were resolved by examining the abstract in which the original text had been used. Words commonly used in most research, including search terms, were selected and excluded by creating a stop word list (e.g., infection, nursing, patient, health, factor, risk, experience, effect, practice, woman, man, evaluation, and impact) [10]. The co-occurrence frequencies of lexemes were calculated and a matrix was generated by applying the co-occurrence frequencies as weights using NetMiner.(4)Text network analysis: To identify the core keywords in the field of infection research, the importance of individual keywords in the network and how close they were to the central position of the network were assessed using the frequency of occurrence, the term frequency–inverse document frequency (TF–IDF) index, and three centrality indices (degree centrality, betweenness centrality, and closeness centrality) [11].


Based on the frequency of occurrence, word clouds based on TF–IDF values were generated to exclude words commonly used in most papers and identify only the important words in each article. The TF is the number of times a particular word appears in a particular document. The IDF is the reciprocal of the frequency of a particular word in all documents [12]. In other words, the TF–IDF index is the weighted frequency obtained by applying a weight to a simple frequency, and it shows how much weight a particular word has in a document. The TF–IDF index is used to evaluate the topic of an article based on the words it contains, since its use of weights enables it to measure relevance, not frequency. Therefore, in the TF–IDF analysis, word counts were replaced with TF–IDF scores across the whole dataset [13]. For this reason, the TF–IDF index indicates the importance of a word more accurately than simple frequency [14]. In order to exclude one-letter words that are difficult to understand, only words with two or more letters were extracted. In addition, only words with a TF–IDF value of 0.70 or more were extracted, thereby excluding words appearing excessively frequently in documents. Through this process, a network composed of a total of 16,169 words was formed.

In centrality analysis, degree centrality shows how many links a node of the network has, and betweenness centrality reflects the degree to which a keyword plays the role of an intermediary in the network. Closeness centrality shows how close a keyword is located to other keywords [15].

(5)Topic modeling: The topics represented by the keywords were identified through topic modeling analysis using the latent Dirichlet allocation (LDA) technique. LDA assumes a document is a mixture of a few unobserved (latent) topics [16]. LDA finds topics that are commonly covered in several documents. LDA collects concurrent words and treats them as a topic. A topic is a set of words that is automatically calculated through computation. LDA allocates words by iteratively executing an algorithm to find the best topic [17]. The number of topics was determined as the silhouette coefficient value through K-means clustering. K-means clustering minimizes the sum of squares of errors within a cluster. The number of topics and values of the LDA parameters are used to select the number of topics with a silhouette coefficient value close to +1 [18]. Besides the top 10 words, those with the highest probability of appearance per topic were visualized as a topic–keyword map, using a topic–word two-mode network. The topic group name was determined by referring to the top words for each topic. When the number of topics was limited to four, the detailed contents of the topics were heterogeneous. For example, when there were four topics, ‘pregnancy’ and ‘HIV’ were grouped together into a single group. Therefore, an attempt was made to determine the number of topics that led to conceptual saturation through several simulations. Through several simulations, a model with 10 topics was determined to be the most reasonable and was adopted.

## 3. Results

### 3.1. Descriptive Findings

In this study, a total of 4854 articles related to infection nursing research from 155 journals were analyzed (with examples of top-ranking journals, including the Journal of the Association of Nurses in AIDS Care, Journal of Advanced Nursing, Heart and Lung, Journal of Clinical Nursing, and Nursing Clinics of North America). With respect to the number of articles according to the year of publication, the first results were two papers on the stigma experienced by trauma patients in 1978. Subsequently, only 511 articles were published in the 1990s. The number of published papers then more than doubled in the next decade, with 1320 papers published in the 2000s and 2948 papers published in the 2010s (Figure 1).

### 3.2. Analysis of the Frequency of Occurrence and TF–IDF

An analysis of keywords in nursing research in terms of the simple frequency of occurrence showed that the keyword with the highest frequency was ‘care’ (239 articles), followed in order by ‘nurse’ (237 articles), ‘HIV’ (237 articles), ‘use’ (125 articles), ‘intervention’ (77 articles), ‘treatment’ (62 articles), ‘hospital’ (60 articles), ‘disease’ (55 articles), ‘knowledge’ (52 articles), and ‘control’ (50 articles) (Table 2).

The keyword with the highest TF–IDF value was ‘readmission’, followed in order by ‘operation’, ‘muscle’, ‘meningitis’, ‘inhibitor’, ‘biopsy’, ‘bath’, ‘aeruginosa’, ‘SARS’, and ‘CD4′ (Table 2).

### 3.3. Centrality Analysis

As a result of centrality analysis using the word network extracted from the abstracts, ‘wound’, ‘injury’, ‘breast’, ‘dressing’, ‘temperature’, ‘drainage’, ‘diabetes’, ‘abscess’, and ‘cleaning’ were identified as the keywords with high values of degree centrality, betweenness centrality, and closeness centrality; hence, they were determined to be influential in the network (Table 3).

### 3.4. Topic Groups Identified by Topic Modeling Analysis

Ten topics were derived through topic modeling analysis (Figure 2). When the number of topics was limited to four, the content of the topics was heterogeneous. Hence, several simulations were performed to determine the number of topics that showed conceptual saturation, and 10 topics were eventually identified. Among the 10 topics, ‘PLWH’ (people living with HIV), and ‘pregnancy’ accounted for a large proportion of the number of articles per topic, and each of the other eight topics comprised around 10%, showing that studies have been conducted on a broad variety of topics. This study examined the sociogram of each group and reviewed the context in which each keyword was used in the abstracts to designate the identified keywords as research topics. Topics 1–10 were as follows: ‘STI’, ‘HAI’, ‘UTI’, ‘PLWH’, ‘HCP’, ‘virus’, ‘pregnancy’, ‘immune topic’, ‘vaccination’, and ‘bacteria’ (Table 4, Figure 2).

## 4. Discussion

This study aimed to identify the core keywords and topics of infection-related research over the past 40 years to better understand the research trends in the field of nursing, and tried to show where nursing research is currently and where it is headed. Our study suggested that a substantial proportion of infection research in nursing has focused on ‘PLWH’, ‘STI’, ‘pregnancy’, and ‘bacteria’. Nursing researchers’ interest in ‘virus’, ‘HAI’, ‘UTI’, and ‘immune topic’ was relatively low, and studies of gastrointestinal infections and community-related research were not identified as significant topics. Our study discovered infection-related research trends by using big data in nursing research to help researchers gain insights into future research. A detailed discussion of these findings is presented below.

Infection-related nursing research has expanded rapidly since the 2000s; the number of published articles more than doubled in the 2000s compared to the previous decade, and more than doubled again in the 2010s. This rapid growth is most likely because the need for the prevention of infections was highlighted due to the worldwide outbreak of emerging infectious disease, as well as the threat posed to patient safety by the rapid increase in hospital-acquired infections (HAIs) with the developments in medical services [19,20,21].

The five keywords with the highest frequency of occurrence were ‘care’, ‘nurse’, ‘HIV’, ‘use’, and ‘intervention’. After excluding ‘care’, ‘use’, and ‘intervention’ as general keywords, the occurrence of ‘nurse’ and ‘HIV’ on this list was notable. It is believed that ‘nurse’ showed a high frequency of occurrence because nurses are involved in infection control in local communities and medical institutions. In addition, HIV was studied overwhelmingly more frequently than other infections. Among the other words included in the top 30 keywords, ‘hospital’, ‘catheter’, ‘wound’, ‘hand’, ‘guideline’, and ‘cancer’ are also remarkable. These keywords show that many infection-related nursing studies have been centered on hospitals, suggesting that the scope of research needs to be expanded to the prevention and control of infections occurring in the community. In addition to ‘HIV’, which was the third ranked keyword and is related to types of infections, ‘catheter’ and ‘wound’ were also among the keywords frequently appearing in nursing research. The high frequency of ‘hand’ is attributed to the importance of hand hygiene for infection prevention. In addition, the keyword ‘guideline’ reflects the fact that the establishment and implementation of guidelines based on a high level of evidence are the main strategies for infection prevention. The keyword ‘cancer’ corresponds to a disease group for which considerable research on infection in nursing has been conducted, due to the fact that cancer patients are at an elevated risk of exposure to infection because their immune system is weakened by cancer treatments.

TF–IDF is a method of presenting important words by excluding keywords that generally appear in many articles. This allows the identification of keywords that are more specifically focused on the topic of infections than those identified by analyzing the raw frequency of occurrence. The word with the highest TF–IDF value was ‘readmission’, followed by ‘operation’, ‘muscle’, ‘meningitis’, and ‘inhibitor’ in descending order. This suggests that infections lead to hospital readmissions, and many studies have been conducted on infection control and surgical wound infections related to surgery [22]. The top 30 words included several keywords related to HAIs, such as ‘CAUTI’, ‘VAP’, and ‘CLABSI’, reflecting topics of interest in nursing research. Keywords with high values of the three centrality indices have many links to other keywords, act as intermediaries, and are located close to other keywords, giving them a great influence on the network. The highly central keywords identified in this study were as follows: ‘wound’, ‘injury’, ‘dressing’, ‘breast’, ‘temperature’, ‘drainage’, ‘diabetes’, and ‘cleaning’. These keywords suggest that overall wound management, including dressing and drainage, is an important topic in the field of infection nursing, and diabetes patients, who are at a high risk of infection and require extra caution in wound management, are major recipients of infection-related nursing care [23,24].

Nonetheless, since bibliometric indicators, such as the frequency of occurrence and centrality indices, have limitations in identifying key themes in a research field, key research topics were derived through topic modeling. In this study, LDA algorithm identified 10 key topics, and are listed in descending order in terms of the group size (determined by number of articles per topic) as follows: ‘PLWH’ (people living with HIV), ‘pregnancy’, ‘STI’, ‘bacteria’, ‘vaccination’, ‘HCP’, ‘HAI’, ‘immune topic’, ‘virus’, and ‘UTI’. It indicated that many studies have been conducted on infection control in the field of nursing. Most of all, this study identified four major topics, such as ‘PLWH’, ‘pregnancy’, ‘STI’, and ‘bacteria’.

Because HIV was a keyword with a high frequency of occurrence, it is unsurprising that PLWH was derived as the key topic with the largest group size. In this regard, the Joint United Nations Program on HIV/AIDS is actively engaging in efforts to end the AIDS epidemic by setting the ‘90-90-90’ target, according to which, by 2020, 90% of HIV-infected people will be aware of their HIV status, 90% of diagnosed HIV-infected patients will receive antiretroviral therapy, and the HIV infection will be suppressed successfully in 90% of patients receiving treatment. In addition to these efforts, the content of relevant nursing research is reflected by the keywords ‘stigma’, ‘transition’, ‘self-care’, and ‘collaboration’, which belonged to the topic of ‘PLWH’. These words illustrate the nature of nursing with the aim of helping HIV-infected patients to overcome social stigma and lead a successful life and perform self-care, including medication therapy [25].

The topic ‘pregnancy’ and sub-keyword ‘condom’ reflect the fact that sexually transmitted infections during pregnancy, which cause vertically transmitted infections from the mother to infant, are considered important in the field of infection research. The Centers for Disease Control and Prevention in the United States recommend that all pregnant women be screened for HIV, syphilis, hepatitis B virus, and chlamydia trachomatis, and that tests for Neisseria gonorrhea and hepatitis C virus should be conducted on high-risk mothers [26]. In addition, infectious diseases related to childbirth are emerging as a major research topic because of the recent outbreak of Zika virus, which leads to microcephaly in infants and threatens the health of pregnant women [27].

‘STI’ was also found to be an emerging research topic in the same context of PLWH research. Among its sub-keywords, ‘violence’, ‘loneliness’, ‘youth’, and ‘threat’ merit ongoing attention in future nursing research. In ‘bacteria topic’, ‘wound’ had a high frequency of occurrence, and its sub-keywords included words related to the management of instruments and devices, such as ‘disinfection’ and ‘sterilization’; words regarding wound management, such as ‘cleaning’ and ‘dressing’; and disinfectants, such as ‘chlorhexidine’. Hospitals accommodate patients who have a high risk of infection due to reduced immunity, as well as people who require surgical, medical, or other treatments. This result shows that the critical role of nurses who come into close contact with these patients is to prevent patients from bacterial infection.

The fact that ‘HCP’ emerged as a key research topic closely related to ‘nurse’, a term with a high frequency of occurrence, reflects the important role of nurses in the prevention and control of infections in the community and medical institutions. This point is supported by sub-keywords, such as ‘staffing’ and ‘nurse’. Medical institutions should secure sufficient infection-related personnel for effective infection control. In addition, there is a growing interest in exposures to infection and subsequent healthcare among healthcare workers. Among the sub-keywords of ‘manpower’ was ‘HCV’, which reflects the increasing frequency of accidental exposure to infections, such as needle-stick injuries, among nurses [28]. Moreover, because of cases involving tuberculosis infections in newborns exposed to infected healthcare workers, the health status of healthcare workers and the importance of a pre-employment physical examination have been emphasized as requirements for infection prevention [29].

‘HAI’ was derived as a key topic because HAIs prolong the length of hospital stay, increase the mortality rate, incur human and economic losses, and cause the spread of bacteria resistant to antibiotics, including multidrug-resistant bacteria, which pose a major threat to public health. In this regard, the importance of detailed nursing management should be emphasized to prevent HAIs in the future. Academic courses on infections are required and the importance of infection control needs to be emphasized in nursing education [22]. In addition, ‘UTI’, reflecting the fact that urinary tract infections can be acquired in the hospital setting, ‘vaccination’, ‘immune topic’, and ‘virus’ were also derived as key topics.

‘Outbreak’, ‘influenza’, ‘SARS’, ‘CAP’, and ‘respiratory’ emerged as key words in the ‘virus’ topic. Currently, the COVID-19 pandemic has brought a number of challenges for nurses. In-depth discussion is needed on the insufficient availability of protective gear and material, uncertainties regarding the trajectory of the disease, which has been a problem in this corona pandemic, and establishment of quarantine measures. Nursing faculty and nursing leaders need to plan for the potential of an extended pandemic response, and for changes that will be needed in a post-COVID world [30].

In this study, while pregnancy, bacteria, and HIV were identified as major research topics, waterborne infections, gastrointestinal infections (cholera, dysentery, and typhoid), and food poisoning were not derived. Because we did not collect information on the countries where studies were conducted, we could not be certain, but we assumed that this finding might have occurred because most of the studies analyzed may have been conducted in developed countries. In the future, further research will be needed to address health inequalities between developed countries and underdeveloped countries. Therefore, further research on gastrointestinal infections, such as waterborne infections and food poisoning, is needed. Moreover, infections occurring in the community were overlooked because most studies were centered on medical institutions.

Although this study only included research conducted through 2017, before the COVID-19 pandemic, ‘virus’ was derived as a main topic. If more recent data were to be included, the proportion of ‘virus’ as a topic would be expected to exceed the value of 8.2% observed in this research. Further research should explore how the situation has changed after the emergence of the COVID-19 pandemic.

### 4.1. Strengths and Limitations

This study used an innovative approach to reveal the trend of infection control studies over the past four decades. This is important as it can inform researchers what further area(s) they should work on to enrich the body of knowledge. Unlike traditional systematic review and meta-analysis, the semantic network analysis tries to present the trend of study which cannot be revealed by systematic review and meta-analysis. Such an analytical approach does enable researchers to examine a study area from another perspective. It is also a trend now that researchers try to use these data mining approaches to reveal the study trend in order to look for area(s) that need further effort to work on. In particular to this study, it clearly identifies the 10 most popular topics being studied, such as STI, HAI, and UTI, in the field of nursing.

Our focus on abstracts rather than the full-text analysis means much data were left unexplored. We feel that abstracts contain more information than titles or keywords, and are considered appropriate as they contain the subject of the entire article. To establish a search strategy that can be objectively verified at the data collection stage, we strategically searched for infection studies with the advice of an external professional, and only those studies in which agreements were reached between all researchers were included in data analysis. Therefore, there is a limitation that some nursing studies related to infection may have been excluded from the analysis.

### 4.2. Implications for Nursing and Health Policy

As a result of examining the trends of infection research in the fields of nursing, it was found that various fields of research were conducted in relation to infectious diseases. Through this study, there is great significance in that it presents a perspective on future research fields by presenting researchers with a relatively undeveloped research field in which research is active and relatively undeveloped. In particular, conducting research on infectious diseases in relation to infection and encouraging research that increases human immune function at the same time is considered to be a preparation for outbreaks related to infectious diseases in the future.

## 5. Conclusions

This study showed that diverse infection research has been conducted on the topics of bloodborne infections, sexually transmitted infections, respiratory infections, urinary tract infections, and bacterial infections. STIs (including HIV), pregnancy, and bacterial infections have been the focus of particularly intense research by nursing researchers. More research on viral infections, urinary tract infections, immune topic, and hospital-acquired infections will be needed.

## Figures and Tables

**Figure 1 ijerph-18-06915-f001:**
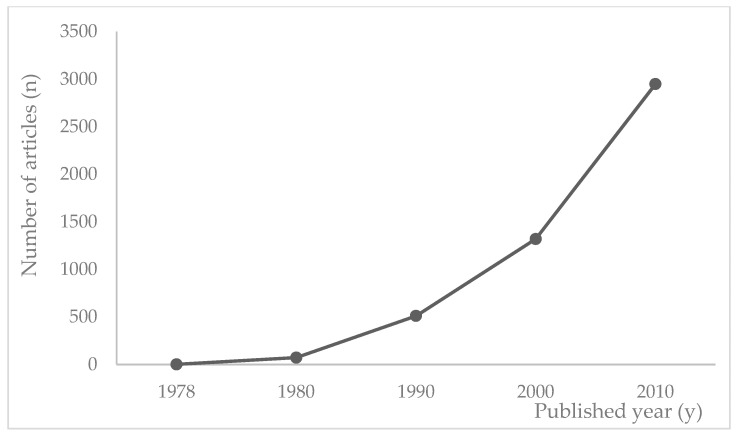
Number of published articles on infection research during 40 years in the nursing field.

**Figure 2 ijerph-18-06915-f002:**
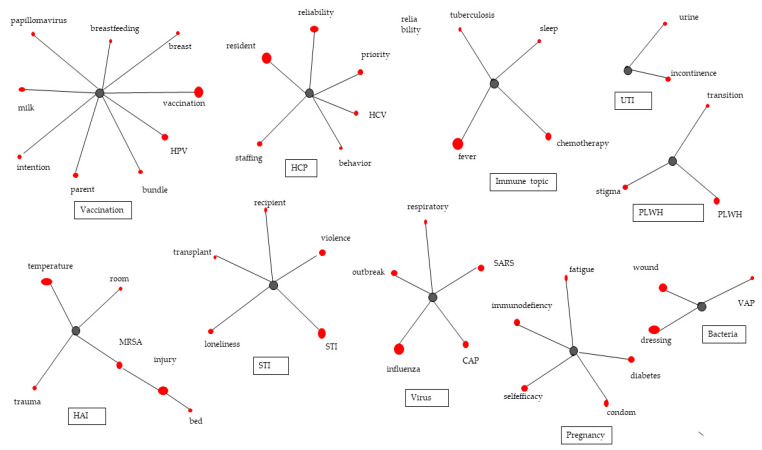
Sociogram of the keywords based on topic modeling. Note. HPV: human papillomavirus; HCP: healthcare professionals; UTI: urinary tract infection; PLWH: people living with HIV; MRSA: methicillin-resistant staphylococcus aureus; HAI: healthcare-associated infection; STI: sexually transmitted infection; SARS: severe acute respiratory syndrome; CAP: community-acquired pneumonia; VAP: ventilator-associated pneumonia.

**Table 1 ijerph-18-06915-t001:** Examples of data cleaning.

Verbatim Text	After Cleaning
Human immunodeficiency virus	HIV
Acquired immune deficiency syndrome	AIDS
Urinary tract infection	UTI
Catheter-associated urinary tract infection	CAUTI
Ventilator-associated pneumonia	VAP
Peripherally inserted central catheter	PICC
Old people, older people, aged person	aged
Nurses, nurses’, nurse’, RN, RNs	nurse
Care, caring, cares	care

Note. HIV: human immunodeficiency virus; AIDS: acquired immune deficiency syndrome; UTI: urinary tract infection; CAUTI: catheter-associated urinary tract infection.; VAP: ventilator-associated pneumonia; PICC: peripherally inserted central catheter; RN: registered nurse.

**Table 2 ijerph-18-06915-t002:** Top 30 lexemes by frequency from infection research in nursing.

Rank	Keyword by Frequency	F	Keyword by TF–IDF	Rank	Keyword by Frequency	F	TF–IDF
1	care	5073	readmission	16	prevention	1164	airway
2	nurse	3910	operation	17	symptom	1141	WHO
3	HIV	4923	muscle	18	diagnosis	1125	intervention
4	use	2156	meningitis	19	management	1084	tumor
5	intervention	1697	inhibitor	20	wound	1079	stressor
6	treatment	1665	biopsy	21	program	1068	hemorrhage
7	hospital	1665	bath	22	adult	1042	fistula
8	disease	1648	aeruginosa	23	complication	1021	exchange
9	knowledge	1632	SARS	24	hand	1007	empowerment
10	control	1567	CD4	25	therapy	1006	cleansing
11	unit	1490	CAUTI	26	guideline	985	amputation
12	catheter	1433	lymphoma	27	change	971	VAP
13	education	1284	dermatitis	28	information	956	PPE
14	child	1284	cytomegalovirus	29	score	949	medicare
15	quality	1219	anemia	30	cancer	934	CLABSI

Note. F: frequency; TF-IDF: term frequency-inverse document frequency; WHO: World Health Organization; HIV: human immunodeficiency virus: SARS: severe acute respiratory syndrome: CD4: cluster of differentiation 4; CAUTI: catheter-associated urinary tract infection; VAP: ventilator-associated pneumonia; PPE: personal protective equipment; CLABSI: central line-associated bloodstream infection.

**Table 3 ijerph-18-06915-t003:** Keywords from network analysis.

Item	Degree Centrality	Betweenness Centrality	Closeness Centrality
1	**wound**	**wound**	**wound**
2	**injury**	**breast**	**breast**
3	fever	fever	**injury**
4	**dressing**	**injury**	burn
5	birth	**dressing**	**drainage**
6	**breast**	management	**dressing**
7	vaccination	**temperature**	prophylaxis
8	**temperature**	self-efficacy	**cleaning**
9	sterilization	**drainage**	bed
10	chemotherapy	condom	fall
11	bladder	burn	maintenance
12	condom	trauma	**abscess**
13	tube	milk	**temperature**
14	milk	**diabetes**	hypertension
15	influenza	hypertension	leg
16	**drainage**	survivor	debridement
17	urine	transplant	trauma
18	transplant	maintenance	disinfection
19	recipient	birth	skin
20	incontinence	bladder	foot
21	**diabetes**	**abscess**	GBS
22	**cleaning**	CVC	swab
23	childhood	vaccination	puncture
24	chest	host	biofilm
25	bed	prophylaxis	hemorrhage
26	room	chemotherapy	cleansing
27	inhibitor	influenza	endocarditis
28	chlorhexidine	acquisition	surface
29	**abscess**	**cleaning**	dehiscence
30	assessment	puncture	**diabetes**

Keywords in bold indicate that the term belongs to all of the top 30 keywords as measured by the centrality measures (degree centrality, betweenness centrality, and closeness centrality). Note. GBS: Group B Streptococcus; CVC: central venous catheter.

**Table 4 ijerph-18-06915-t004:** Topic group and high ranking keywords. (n = 4854).

Group	Topic Name	N (%)	1st Keyword	Probability	2nd Keyword	Probability	3rd Keyword	Probability	4th Keyword	Probability	5th Keyword	Probability
Topic 1	STI	502 (10.8)	violence	0.151	transplant	0.051	STI	0.037	loneliness	0.029	recipient	0.026
Topic 2	HAI	414 (8.9)	injury	0.153	temperature	0.049	MRSA	0.024	bed	0.024	room	0.016
Topic 3	UTI	378 (8.2)	incontinence	0.031	urine	0.029	infusion	0.028	comfort	0.025	tube	0.025
Topic 4	PLWH	641 (13.8)	stigma	0.067	PLWH	0.043	transition	0.037	specialist	0.032	network	0.029
Topic 5	HCP	445 (9.6)	resident	0.074	behavior	0.066	staffing	0.05	HCV	0.039	priority	0.032
Topic 6	Virus	381 (8.2)	influenza	0.152	respiratory	0.101	outbreak	0.024	SARS	0.021	CAP	0.02
Topic 7	Pregnancy	515 (11.1)	birth	0.108	condom	0.051	fatigue	0.036	diabetes	0.03	immunodeficiency	0.029
Topic 8	Immune topic	409 (8.8)	fever	0.057	caregiver	0.05	chemotherapy	0.042	tuberculosis	0.041	sleep	0.026
Topic 9	Vaccination	457 (9.9)	vaccination	0.056	HPV	0.025	parent	0.022	breast	0.02	breastfeeding	0.017
Topic 10	Bacteria	492 (10.6)	wound	0.05	dressing	0.037	VAP	0.026	chlorhexidine	0.022	surface	0.022

Note. STI: sexually transmitted infection; HAI: healthcare-associated infection; MRSA: methicillin-resistant staphylococcus aureus; UTI: urinary tract infection; PLWH: people living with HIV; HCP: healthcare professionals; HCV: hepatitis C virus; SARS: severe acute respiratory syndrome; CAP: community-acquired pneumonia; HPV: human papillomavirus; VAP: ventilator-associated pneumonia.

## Data Availability

The data presented in this study are available on request from the corresponding author.

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
