# Peer review of "Trends in Nursing Research on Infections: Semantic Network Analysis and Topic Modeling"

_ijerph, 2021, doi:10.3390/ijerph18136915_

Round 1

Reviewer 1 Report

The article proposes methods for analyzing research topics on infections in nursing and public health for 40 years (1978-2018), aiming to understand trends in strategies used in nursing to fight infections. Data collection (articles) and qualitative and quantitative analysis using statistical methods correlate terms in the area. Some visualizations and data tables are used to present the results obtained, and finally, descriptive scenarios of the relationship between the keywords obtained are presented.

The materials and methods presented highlight the data collection carried out on the web of science platform limited to the period from 1978 to 04/2018 and the work of researchers to identify keywords, terms, and review of specialized literature, which stands out in this context. There is a lack of justifications or references for establishing the list of the most relevant keywords, metrics, cut-off factors values for keywords, techniques adopted, and among other manual decisions throughout the process.

The selection of works for analysis (7802) was based on exclusion factors applied by a group of 7 researchers who jointly debated the contents and results, reaching a consensus of 4854 works for analysis, thus generating a co-occurrence matrix of the extracted keywords. The study applies a network text analysis based on the frequency of terms, the index of the inverse frequency of the document, and the centrality index to support analysis. Some concepts presented need references.

The visualization shown in figure 1 is recommended to insert labels and legends on the axes. In table 2, it is recommended to add the value of the frequency associated with the Lexeme. Table 3 presents a set of terms ranked based on centrality measures; keywords in bold have been highlighted and need to be better explained their occurrence between the three metrics.

Specifically, topic 3.4 emphasizes concepts already seen in another section. Table 4 presents the term "n" equals 4634, understanding that “n” corresponds to the number of selected works, it is identified that it is different from the 4854 previously presented, it is recommended to add the value of the occurrences for the other words related to each topic. In figure 2, it is recommended to add the visual variable size for the sociogram nodes, relating the node to the frequency of occurrence of the term.

My suggestion is to improve the text mainly in the methodological part, explaining in a bit more detail the manual decisions that future work can reproduce or extend the actual research for the years 2018-2021 example.  

Author Response

Review1

Dear Reviewer,

We are grateful for your consideration of this manuscript, and we also very much appreciate your suggestions, which have been very helpful in improving the manuscript. We also thank the reviewers for their careful reading of our text.

All the comments we received on this study have been taken into account in improving the quality of the article, and we present our reply to each of them separately. With regard to some of the suggestions, we would note the following:

  1. The materials and methods presented highlight the data collection carried out on the web of science platform limited to the period from 1978 to 04/2018 and the work of researchers to identify keywords, terms, and review of specialized literature, which stands out in this context. There is a lack of justifications or references for establishing the list of the most relevant keywords, metrics, cut-off factors values for keywords, techniques adopted, and among other manual decisions throughout the process.

-We made up for the lack of explanations throughout the study method and added references for concept in text network analysis (line 74~170).

  1. The study applies a network text analysis based on the frequency of terms, the index of the inverse frequency of the document, and the centrality index to support analysis. Some concepts presented need references.

–We added references for concepts presented (line 136, 143)

  1. The visualization shown in figure 1 is recommended to insert labels and legends on the axes. In table 2, it is recommended to add the value of the frequency associated with the Lexeme. Table 3 presents a set of terms ranked based on centrality measures; keywords in bold have been highlighted and need to be better explained their occurrence between the three metrics.

- As your comment, we inserted label and legends in figure. We explained what keywords in bold means.

  1. Specifically, topic 3.4 emphasizes concepts already seen in another section. Table 4 presents the term "n" equals 4634, understanding that “n” corresponds to the number of selected works, it is identified that it is different from the 4854 previously presented, it is recommended to add the value of the occurrences for the other words related to each topic. In figure 2, it is recommended to add the visual variable size for the sociogram nodes, relating the node to the frequency of occurrence of the term.

- We corrected typographical error and added probability of the occurrences for keywords. We modified the figure 2 as your comment.

Reviewer 2 Report

This is an interesting piece of work. It would be more beneficial to add a write-up on how this work is relevant to the current COVID-19 situation. A discussion of the findings of the network analysis is needed.

Author Response

Review2

Dear Reviewer,

We are grateful for your consideration of this manuscript, and we also very much appreciate your suggestions, which have been very helpful in improving the manuscript. We also thank the reviewers for their careful reading of our text.

All the comments we received on this study have been taken into account in improving the quality of the article, and we present our reply to each of them separately. With regard to some of the suggestions, we would note the following:

  1. This is an interesting piece of work. It would be more beneficial to add a write-up on how this work is relevant to the current COVID-19 situation. A discussion of the findings of the network analysis is needed.

-We added the findings of the network analysis and Covid 19 pandemic issues in a discussion part(line 55~63, line 115~121).

Nonetheless, since bibliometric indicators, such as the frequency of occurrence and centrality indices, have limitations in identifying key themes in a research field, key research topics were derived through topic modeling. In this study, LDA algorithm identified 10 key topics, and are listed in descending order in terms of the group size (determined by number of articles per topic) as follows: ‘PLWH’ (people living with HIV), ‘pregnancy’, ‘STI’, ‘bacteria’, ‘vaccination’, ‘HCP’, ‘HAI’, ‘immune topic’, ‘virus’, and ‘UTI’. It indicated that many studies have been conducted on infection control in the field of nursing. Most of all, this study identified 4 major topics such as ‘PLWH’, ‘pregnancy’, ‘STI’. ‘Bacteria’.

‘Outbreak’, ‘influenza’, ‘SARS’, ‘CAP’, and ‘respiratory’ emerged as key words in the ‘virus’ topic. Currently, the Covid-19 pandemic has brought a number of challenges for nurses. In-depth discussion is needed on the insufficient availability of protective gear and material, uncertainties regarding the trajectory of the disease, which has been a problem in this corona pandemic, and establishment of quarantine measures. Nursing faculty and nursing leaders need to plan for the potential of an extended pandemic response, and for changes that will be needed in a post-COVID world [30].

Reviewer 3 Report

Thank you for your interesting manuscript. A few questions have surfaced. 

  • The Covid-19 pandemic has brought a number of challenges for nurses. However, most of these challenges are due to exceedingly high workload, changes in work-related laws and regulations, insufficient availability of protective gear and material, uncertainties regarding the trajectory of the disease and other many unknowns. In the manuscript, a link is made with the HIV/AIDS epidemic. However, no argument is provided to explain this linkage. 
  • Covid-19 is transmitted predominantly via aerosols. Hence, the link with HIV/AIDS is questionable as this is a sexually transmitted disease for the most part. Please provide additional explanations. 
  • Nurses are supposed to be up to date on hygiene and related measures. The point made here suggests that nurses are not sufficiently trained or do not possess enough knowledge to act in a meaningful and evidence-based way. Please explain.
  • The research question and the subsequent study, therefore, are not readily explained. It is necessary to review the argument in order to improve the conducted study. 
  • Please provide clear and concrete recommendations for clinical practice. 

Sincerely, your reviewer. 

Author Response

Review 3

Dear Reviewer,

We are grateful for your consideration of this manuscript, and we also very much appreciate your suggestions, which have been very helpful in improving the manuscript. We also thank the reviewers for their careful reading of our text.

All the comments we received on this study have been taken into account in improving the quality of the article, and we present our reply to each of them separately. With regard to some of the suggestions, we would note the following:

  1. The Covid-19 pandemic has brought a number of challenges for nurses. However, most of these challenges are due to exceedingly high workload, changes in work-related laws and regulations, insufficient availability of protective gear and material, uncertainties regarding the trajectory of the disease and other many unknowns. In the manuscript, a link is made with the HIV/AIDS epidemic. However, no argument is provided to explain this linkage.

-We added Covid 19 pandemic issues in a discussion part line 115~121).

‘Outbreak’, ‘influenza’, ‘SARS’, ‘CAP’, and ‘respiratory’ emerged as key words in the ‘virus’ topic. Currently, the Covid-19 pandemic has brought a number of challenges for nurses. In-depth discussion is needed on the insufficient availability of protective gear and material, uncertainties regarding the trajectory of the disease, which has been a problem in this corona pandemic, and establishment of quarantine measures. Nursing faculty and nursing leaders need to plan for the potential of an extended pandemic response, and for changes that will be needed in a post-COVID world [30].

  1. Covid-19 is transmitted predominantly via aerosols. Hence, the link with HIV/AIDS is questionable as this is a sexually transmitted disease for the most part. Please provide additional explanations. 

-This study aimed to identify the trend of infection studies in nursing using a semantic network analysis. Through study results, topics of infection research in nursing were identified, and 'PLWH' and 'STI' was identified as major topic. Accordingly, research trends related to 'PLWH', 'HIV' and 'infection' were described in the discussion.

  1. Nurses are supposed to be up to date on hygiene and related measures. The point made here suggests that nurses are not sufficiently trained or do not possess enough knowledge to act in a meaningful and evidence-based way. Please explain.

- In this study, 10 key topics were derived, and are listed in descending order in terms of the group size (determined by number of articles per topic) as follows: ‘PLWH’ (people living with HIV), ‘pregnancy’, ‘STI’, ‘bacteria’, ‘vaccination’, ‘HCP’, ‘HAI’, immune topic’, ‘virus’, and ‘UTI’. It did not mean that nurses are not sufficiently trained or do not possess enough knowledge to act in a meaningful and evidence-based way. We presented the overview for research trend of infection –related studies.

  1. The research question and the subsequent study, therefore, are not readily explained. It is necessary to review the argument in order to improve the conducted study. Please provide clear and concrete recommendations for clinical practice. 

Research question was to describe research trend of infection studies in nursing. This study showed that there were 10 most popular topics in infection-related nursing studies. We added concrete recommendations for education and clinical practice. In the strength and implication section, we highlighted research question and subsequent study.

‘Outbreak’, ‘influenza’, ‘SARS’, ‘CAP’, and ‘respiratory’ emerged as key words in the ‘virus’ topic. Currently, the Covid-19 pandemic has brought a number of challenges for nurses. In-depth discussion is needed on the insufficient availability of protective gear and material, uncertainties regarding the trajectory of the disease, which has been a problem in this corona pandemic, and establishment of quarantine measures. Nursing faculty and nursing leaders need to plan for the potential of an extended pandemic response, and for changes that will be needed in a post-COVID world. s a result of examining the trends of infection research in the fields of nursing and health, it was found that various fields of research were conducted in relation to infectious diseases. Through this study, it has great significance in that it presents a perspective on future research fields by presenting researchers with a relatively undeveloped research field in which research is active and relatively undeveloped. In particular, conducting research on infectious diseases in relation to infection and encouraging research that increases human immune function at the same time is considered to be a preparation for outbreaks related to infectious diseases in the future.

4.2 Implications for Nursing and Health Policy

As a result of examining the trends of infection research in the fields of nursing, it was found that various fields of research were conducted in relation to infectious diseases. Through this study, it has great significance in that it presents a perspective on future research fields by presenting researchers with a relatively undeveloped research field in which research is active and relatively undeveloped. In particular, conducting research on infectious diseases in relation to infection and encouraging research that increases human immune function at the same time is considered to be a preparation for outbreaks related to infectious diseases in the future.

Reviewer 4 Report

The study is well-presented. The analytical method is particularly novel and will be of interest to international readers. 

The study aimed to identify the trend of infection control studies in
nursing using a semantic network analysis. It identified the core
keywords and topics of infection-related research over the past 40 years
which is the first study providing such info to readers of interest. The
authors have used an innovative approach to reveal the trend of
infection control studies over the past four decades. It is important as
it can inform researchers what further area(s) they should work on to
enrich the body of knowledge. Unlike traditional systematic review and
meta-analysis, the semantic network analysis tries to present the trend
of study which cannot be revealed by systematic review and
meta-analysis. Such analytical approach does enable researchers to
examine a study area from another perspective. It is also a trend now
that researchers try to use these data mining approaches to reveal the
study trend in order to look for area(s) that need further effort to
work on.

In particular to this study, it clearly explains what semantic network
analysis is and how it helps to dig out the required information
(section 2.2). in the result section, it clearly tell the 10 most
popular topics being studied such as STI, HAI, UTI. To further improve
the article, I would suggest the authors to add a paragraph summarizing
the major implication of the study findings to nursing and healthcare to
facilitate the understanding of the findings given this analytical
approach is new to us and further expand the paragraph about limitation
(the 2nd last para).

Author Response

Review4

Dear Reviewer,

We are grateful for your consideration of this manuscript, and we also very much appreciate your suggestions, which have been very helpful in improving the manuscript. We also thank the reviewers for their careful reading of our text.

All the comments we received on this study have been taken into account in improving the quality of the article, and we present our reply to each of them separately. With regard to some of the suggestions, we would note the following:

  1. In particular to this study, it clearly explains what semantic network analysis is and how it helps to dig out the required information (section 2.2). in the result section, it clearly tell the 10 most popular topics being studied such as STI, HAI, UTI. To further improve the article, I would suggest the authors to add a paragraph summarizing he major implication of the study findings to nursing and healthcare to facilitate the understanding of the findings given this analytical approach is new to us and further expand the paragraph about limitation (the 2nd last para).

-We added ‘strength and implication’ section and the paragraph about study limitation.

4.1 Strengths and Limitations

This study have used an innovative approach to reveal the trend of infection control studies over the past four decades. It is important as it can inform researchers what further area(s) they should work on to enrich the body of knowledge. Unlike traditional systematic review and meta-analysis, the semantic network analysis tries to present the trend of study which cannot be revealed by systematic review and meta-analysis. Such analytical approach does enable researchers to examine a study area from another perspective. It is also a trend now that researchers try to use these data mining approaches to reveal the study trend in order to look for area(s) that need further effort to work on. In particular to this study, it clearly tell the 10 most popular topics being studied such as STI, HAI, UTI in the field of Nursing.

Our focus on abstracts rather than the full-text analysis means much data was left unexplored. We feel that abstracts contain more information than titles or keywords, and are considered appropriate as they contain the subject of the entire article. To establish a search strategy that can be objectively verified at the data collection stage, we strategically searched for infection studies with the advice of an external professional, and only those studies in which agreements were reached between all researchers were included in data analysis. Therefore, there is a limitation that some nursing studies related to infection may have been excluded from the analysis.

4.2 Implications for Nursing and Health Policy

As a result of examining the trends of infection research in the fields of nursing, it was found that various fields of research were conducted in relation to infectious diseases. Through this study, it has great significance in that it presents a perspective on future research fields by presenting researchers with a relatively undeveloped research field in which research is active and relatively undeveloped. In particular, conducting research on infectious diseases in relation to infection and encouraging research that increases human immune function at the same time is considered to be a preparation for outbreaks related to infectious diseases in the future.
